# Maintaining immunogenicity of blood stage and sexual stage subunit malaria vaccines when formulated in combination

**Elizabeth M. Parzych[1], Kazutoyo Miura[2], Carole A. Long[1,2], James M. Burns, Jr.**[1] *

**1** Department of Microbiology and Immunology, Center for Molecular Parasitology, Drexel University College of Medicine, Philadelphia, Pennsylvania, United States of America, **2** Malaria Immunology Section, Laboratory of Malaria and Vector Research, National Institute of Allergy and Infectious Diseases, National Institutes of Health, Rockville, Maryland, United States of America

* jmb53@drexel.edu

**Data Availability Statement:** All relevant data are within the manuscript and its Supporting Information files.

## Abstract

### Background

Eradication of *Plasmodium falciparum* malaria will likely require a multivalent vaccine, but the development of a highly efficacious subunit-based formulation has been challenging. We previously showed that production and immunogenicity of two leading vaccine targets, *Pf*MSP1$_{19}$ (blood-stage) and *Pf*s25 (sexual stage), could be enhanced upon genetic fusion to merozoite surface protein 8 (*Pf*MSP8). Here, we sought to optimize a *Pf*s25-based formulation for use in combination with r*Pf*MSP1/8 with the goal of maintaining the immunogenicity of each subunit.

### Methods

Comparative mouse studies were conducted to assess the effects of adjuvant selection (Alhydrogel vs. glucopyranosyl lipid adjuvant-stable emulsion (GLA-SE)) and antigen dose (2.5 vs. 0.5 μg) on the induction of anti-*Pf*s25 immune responses. The antibody response (magnitude, IgG subclass profile, and transmission-reducing activity (TRA)) and cellular responses (proliferation, cytokine production) generated in response to each formulation were assessed. Similarly, immunogenicity of a bivalent vaccine containing r*Pf*MSP1/8 and r*Pf*s25/8 was evaluated.

### Results

Alum-based formulations elicited strong and comparable humoral and cellular responses regardless of antigen form (unfused r*Pf*s25 or chimeric r*Pf*s25/8) or dose. In contrast, GLA-SE based formulations elicited differential responses as a function of both parameters, with 2.5 μg of r*Pf*s25/8 inducing the highest titers of functional anti-*Pf*s25 antibodies. Based on these data, chimeric r*Pf*s25/8 was selected and tested in a bivalent formulation with r*Pf*MSP1/8. Strong antibody titers against *Pf*s25 and *Pf*MSP1$_{19}$ domains were induced with GLA-SE based formulations, with no indication of antigenic competition.

**Funding:** This work was supported by NIH-NIAID Grant AI114292 (JMB), the Intramural Program of NIH-NIAID (CAL), and PATH's Malaria Vaccine Initiative (via support of the SMFA Reference Laboratory at the Laboratory of Malaria and Vector Research, NIAID, NIH). The funders had no role in study design, data collection and analysis, decision to publish or preparation of the manuscript.

**Competing interests:** I have read the journal's policy and the authors of this manuscript have the following competing interests: JMB is an inventor listed on US Patent No. 7,931,908 entitled "Chimeric MSP-Based Malaria Vaccine". This does not alter our adherence to PLOS ONE policies on sharing data and materials.

## Conclusions

We were able to generate an immunogenic bivalent vaccine designed to target multiple parasite stages that could reduce both clinical disease and parasite transmission. The use of the same *Pf*MSP8 carrier for two different vaccine components was effective in this bivalent formulation. As such, the incorporation of additional protective targets fused to the *Pf*MSP8 carrier into the formulation should be feasible, further broadening the protective response.

## Introduction

Significant strides have been made in the control and treatment of malaria since the year 2000. However, there has been a rise in drug-resistant parasites and insecticide-resistant mosquitos, and progress towards elimination has stalled over recent years. Development of additional tools, including highly efficacious vaccines, would greatly aid efforts to further decrease clinical disease and mortality due to *Plasmodium falciparum*. Considering the suboptimal protection afforded by single antigen vaccines such as RTS,S [1–3], it is likely that induction of broad responses against multiple targets will be required to achieve adequate efficacy. While the ideal vaccine would induce sterilizing immunity, a more attainable, yet still impactful goal, may be the development of a multistage vaccine capable of reducing both the severity of clinical disease and parasite transmission rates.

One strategy being pursued for the rational development of a multivalent subunit malaria vaccine requires the production of high-quality and potent recombinant immunogens that can be successfully combined into a single formulation while adequately maintaining the protective effect of each component. These have been challenges for the field, as many protective targets are structurally complex and difficult to produce properly in recombinant form. Furthermore, antigenic competition has been observed with various formulations that incorporated multiple pre-erythrocytic and/or blood-stage antigens [4–6]. We developed a strategy to help facilitate this process and address the issues of vaccine production, folding and immunogenicity while minimizing antigenic competition, through the use of merozoite surface protein 8 (*Pf*MSP8) as a malaria-specific carrier protein.

Antibodies directed against conformational epitopes within the C-terminal epidermal growth factor-like domains of *P. falciparum* merozoite surface protein 1 (*Pf*MSP1) are highly protective in rodent and non-human primate models of malaria [7–14]. However, in clinical trials of *Pf*MSP1$_{42}$, efficacy was limited due, in part, to suboptimal immunogenicity and epitope polymorphism [15–21]. Our early studies in the *P. yoelii* rodent model pointed to the potential of MSP8 as a vaccine carrier to avoid antigenic competition, to enhance the production of *Py*MSP1$_{19}$-specific antibodies and to provide solid [22] and durable [23] protection against lethal *P. yoelii* malaria. Therefore, we tested the utility of this approach for *P. falciparum*. *Pf*MSP8 was engineered to be highly expressed, properly folded and easily purified using an *E. coli* expression system [24]. To assess the ability of *Pf*MSP8 to enhance the production, folding and immunogenicity of *Pf*MSP1$_{19}$, a chimeric antigen containing r*Pf*MSP1$_{19}$ genetically fused to the N-terminus *Pf*MSP8 was generated [25]. The resulting fusion protein, r*Pf*MSP1/8, i) was expressed and purified in high yield, bearing proper conformation of the *Pf*MSP1$_{19}$ domain, ii) induced a predominant *Pf*MSP8-specific T cell response, iii) elicited high titers of antigen-specific antibodies in inbred and outbred mice, rabbits and non-human primates, which were cross-reactive with *Pf*MSP1$_{19}$ from the FVO and 3D7 strains of *P. falciparum*, and iv) could be formulated with diverse adjuvants to stimulate production of anti-

*Pf*MSP1$_{19}$ antibodies that potently inhibited the *in vitro* growth of *P. falciparum* blood-stage parasites. Using a similar strategy, we have also reported success utilizing *Pf*MSP8 as a carrier for a second blood-stage target, *P. falciparum* merozoite surface protein 2 (*Pf*MSP2), to elicit antibodies that opsonize merozoites for phagocytosis [26].

*Pf*s25 is a highly conserved, 25 kDa glycosylphosphatidylinositol (GPI) anchored surface protein expressed exclusively during the sexual stages of the parasite life cycle within the mosquito midgut [27]. It is well established that vaccine-induced antibodies directed against conformational epitopes within the four EGF-like domains of *Pf*s25 are able to block sexual stage development within the vector, effectively preventing parasite transmission [28–32]. This induction of transmission-blocking immunity has been demonstrated in mouse models, non-human primates and human subjects. However, similar to *Pf*MSP1, it has been difficult to produce sufficient quantities of high quality recombinant *Pf*s25 bearing proper conformation using common expression systems. Thus far, clinical trials conducted on *Pf*s25-based candidates have resulted in suboptimal immunogenicity and durability of vaccine induced responses [33–36]. To begin to address these issues, we produced a chimeric r*Pf*s25-*Pf*MSP8 fusion protein as well as unfused, mature r*Pf*s25 [37]. r*Pf*s25 was purified with a modest yield but required denaturation and renaturation procedures to obtain the correct conformation. In contrast, r*Pf*s25/8 was purified in higher yield without the need for refolding. Both antigens were immunogenic in rabbits, inducing IgG that bound native, macrogamete-associated *Pf*s25 and exhibited potent transmission-reducing activity in a standard membrane feeding assay (SMFA).

Here, we sought to systematically assess the relative immunogenicity of these *Pf*s25-based vaccines as a function of several formulation parameters including adjuvant selection and antigen dose, with the ultimate goal of selecting an optimized *Pf*s25-based antigen for incorporation into a multivalent vaccine. We tested the influence of two distinct human-compatible adjuvants on the anti-*Pf*s25 responses. Alhydrogel (Alum), a safe and widely used adjuvant for childhood vaccines, has been shown to enhance humoral immunity and skew immune responses toward a Th$_2$ profile with production of IL-5 and IgG1 antibodies [38, 39]. In contrast, GLA-SE is a two-component adjuvant that contains glucopyranosyl lipid adjuvant (GLA), a synthetic TLR4 agonist, in a stable squalene-in-water emulsion (SE). GLA-SE shifts responses toward a Th$_1$ profile characterized by increased production of IFNγ and TNFα with a more diverse IgG subclass profile featuring increased levels of the cytophilic IgG2a/c in mice [40]. As a next step in building a multistage vaccine, the *Pf*s25/8 and *Pf*MSP1/8 vaccines were tested in combination to i) assess the potential for antigenic competition, ii) select an optimal adjuvant for the bivalent formulation, and iii) determine the impact of concurrent immunization with two subunit vaccines fused to the same carrier protein.

## Materials and methods

### Mice and immunizations

Five-week-old, male CB6F1/J mice (BALB/c x C57BL/6) or male and female outbred CD1 mice were obtained from The Jackson Laboratory and Charles River Laboratories, respectively. Mice were maintained in the Animal Care Facility of Drexel College of Medicine under specific-pathogen-free conditions. All animal studies were designed, reviewed, approved and conducted in accordance with the Institutional Animal Care and Use Committee of Drexel University College of Medicine (protocol # 20308). For comparative immunogenicity studies in CB6F1/J mice, groups (n = 5) were immunized with 0.5 μg/dose (low) or 2.5 μg/dose (high) of purified r*Pf*s25, r*Pf*s25/8, r*Pf*MSP8 or an admixture of r*Pf*s25 + r*Pf*MSP8 (0.5 μg or 2.5 μg of each antigen/dose). Production and purification of recombinant antigens have been previously

reported [37]. Antigens were formulated in either 2% Alhydrogel adjuvant (500 µg/dose; Invi-voGen, San Diego, CA) or GLA-SE (5 µg/dose, Infectious Disease Research Institute, Seattle, WA). Additional control groups received adjuvant alone. For the bivalent vaccine study, groups of CD1 mice (n = 10; 5 male and 5 female) were immunized subcutaneously with 2.5 µg/dose of purified r*Pf*s25/8, r*Pf*MSP1/8, an admixture of r*Pf*MSP1/8 + r*Pf*s25/8 (2.5 µg of each antigen/dose) or adjuvant alone. Antigens were formulated, as above, with Alum or GLA-SE as adjuvant. For assessment of antibody responses, mice were immunized subcutaneously, three times at 4-week intervals. Sera samples were collected three weeks following the first two immunizations and 4 weeks following the final immunization. For assessment of T cell responses, mice were immunized subcutaneously three times at 4-week intervals. Following an 8–10 week rest, mice received an additional boost by intraperitoneal (i.p.) injection to increase trafficking of antigen-specific T cells to the spleen. Splenocytes were harvested 2 weeks following the i.p. boost.

### Antigen-specific T cell analysis

**Splenocyte preparation.** Harvested spleens were processed into single cell suspensions in sterile complete medium consisting of RPMI 1640 (Sigma-Aldrich, St. Louis, MO) supplemented with 2 mM L- glutamine, 0.5 mM sodium pyruvate, 50 µM 2-mercaptoethanol, 1X streptomycin/penicillin (Corning Costar Corporation, Cambridge, MA), 10 µg/ml of Poly-myxin B (Sigma-Aldrich) and 10% heat-inactivated Benchmark™ fetal bovine serum (Gemini Bio Products, Sacramento, CA). Cellular debris was removed from suspensions by filtration through Falcon 70 µm cell strainers (Thermo Fisher Scientific). RBCs were lysed using ACK lysis buffer (Thermo Fisher Scientific), and the quantification of viable splenocytes was determined by microscopy following trypan blue staining (Thermo Fisher Scientific).

**T cell proliferation assay.** To measure antigen-specific proliferative responses, splenocytes (5 mice/group) were plated in 96-well round-bottomed Falcon plates (Thermo Fisher Scientific) at a concentration of $2 \times 10^5$ cells/well. Cells from each mouse were stimulated in triplicate in RMPI complete medium containing 10 µg/ml of r*Pf*s25, r*Pf*s25/8, or r*Pf*MSP8 antigens. Additional sets of wells from each mouse were stimulated in triplicate with Concanavalin A (Sigma-Aldrich; 1 µg/ml) or left unstimulated to serve as positive and negative controls, respectively. Plates were incubated at 37˚C in 5% $CO_2$ for 96 hours, and pulsed with methyl [³H]-thymidine (1 µCi/well; 70–90 Ci/mmol; PerkinElmer, Inc., Waltham, MA) for the final 18 hours. Cells were harvested onto glass fiber filters using an automatic cell harvester (Perki-nElmer, Inc.). Incorporation of [³H]-thymidine was quantified by liquid scintillation counting (PerkinElmer, Inc). The stimulation indices were calculated for each animal as the mean counts per minute of each stimulated condition divided by the mean counts per minute of the corresponding unstimulated condition.

**Cytokine production.** For the quantification of secreted cytokines induced by antigen-specific stimulation, splenocytes were plated in 96-well round-bottomed Falcon plates at a concentration of $5 \times 10^5$ cells/well and stimulated as described above for 96 hours. Culture supernatants were transferred to new plates and stored at -80˚C. Custom magnetic Luminex® assay kits (R&D Systems, Minneapolis, MN) were used for the quantification of IL-2, IL-4, IL-5, TNFα and IFNγ in cell supernatants according to the manufacturer's protocol, utilizing a Luminex 200 analyzer and xPONENT3.1 software. Based on a standard curve, concentrations (pg/ml) of each analyte were calculated for all samples and final, antigen-specific concentrations were determined by subtracting out the background levels in corresponding unstimulated conditions.

## Determination of antigen-specific antibody titers

**Enzyme-linked immunosorbent assay (ELISA).** Sera collected from all experimental and control mice following each immunization were analyzed for antigen-specific IgG by ELISA as previously described [25]. Briefly, plates coated with 0.25 μg/well of r*Pf*s25, r*Pf*s25/8 or r*Pf*MSP8 were incubated with two-fold dilutions of mouse sera for two hours at room temperature. Bound antibodies were detected by HRP-conjugated rabbit anti-mouse IgG (0.08 μg/ml; ThermoFisher Scientific) and ABTS [2,2'-azinobis(3-ethylbenzothiazoline-6-sulfonic acid)] as substrate. $A_{405}$ values between 0.1 and 1 were plotted and titers were calculated as the reciprocal of the dilution yielding an $A_{405}$ of 0.5. A high titer pool of sera obtained from r*Pf*s25/8-immunized mice was included on every plate to normalize data between plates.

**IgG subclass profiles of antigen-specific antibodies.** To determine the IgG subclass profiles of antigen-specific IgG, tertiary immunization serum from each mouse was titered, as described above, in wells coated with r*Pf*s25/8. Bound antibodies were detected by HRP-conjugated rabbit anti-mouse IgG specific for subtypes IgG1, IgG2a, IgG2b, IgG2c and IgG3 (Southern Biotech, Inc., Birmingham, AL.) followed by ABTS substrate. To generate a standard curve, each plate included wells coated with 2-fold dilutions of subtype-specific mouse myeloma immunoglobulin at known concentrations. Here, IgG subclass quantities in sera are reported as units/ml (U/ml) where 1 U/ml is equivalent to the signal obtained with 1 μg/ml of purified myeloma protein.

## Standard membrane feeding assay

The transmission-reducing activity (TRA) of IgG antibodies induced by each vaccine formulation was measured by a Standard Membrane Feeding Assay (SMFA) using cultured *P. falciparum* NF54 gametocytes and *Anopheles stephensi* mosquitoes, as previously described [41]. Pools of protein G-purified, vaccine-induced IgG (750 μg/ml) were mixed with stage V gametocytes and fed to *A. stephensi* mosquitoes through a membrane feeding apparatus. Mosquitoes were kept for 8 days prior to dissection to quantify midgut oocysts. Percent inhibition of mean oocyst intensity was calculated relative to adjuvant control IgG. The best estimate of % inhibition in mean oocyst density (% TRA), the 95% confidence interval, and the *p*-value (whether the observed %TRA is significantly different from no inhibition) of each test sample were calculated using a zero-inflated negative binomial model [42].

## Statistical analysis

All statistical analyses conducted in this study were nonparametric. To assess T cell responses (proliferation and cytokine production) in antigen-immunized groups relative to the corresponding control group, a Kruskal-Wallis test was conducted. To assess boosting of antigen-specific IgG responses within the same animals following each immunization, a Friedman's test for multiple repeated samples followed by a Dunn's post hoc test was utilized. For analysis of final anti-*Pf*s25 titers induced by immunization with r*Pf*s25, r*Pf*s25 + r*Pf*MSP8, or r*Pf*s25/8 in the different dose and adjuvant formulations, a Kruskal-Wallis test followed by a Dunn's post hoc test was performed. Instances in which two unrelated groups were directly compared, Mann-Whitney *U* tests were used. In all cases, differences with a probability (*p*) value of <0.05 were considered significant.

## Results

The relative immunogenicity of the two recombinant *Pf*s25-based vaccines, unfused r*Pf*s25 and chimeric r*Pf*s25/8, was compared as a function of various vaccine parameters including

antigen form (r*Pf*s25, r*Pf*s25/8 or an admixture of r*Pf*s25 + r*Pf*MSP8), antigen dose and adjuvant selection. Antibody responses induced by each vaccine formulation were measured with respect to magnitude, specificity, IgG subclass and functionality. In addition, the phenotype and specificity of vaccine-induced T cells were assessed.

### *Pf*s25 and *Pf*MSP8 domains elicit antigen-specific T cell responses

The domain specificity of T cell responses elicited by immunization with r*Pf*s25 and r*Pf*s25/8 antigens was determined. To this end, splenocytes from immunized animals were harvested and stimulated *in vitro* with r*Pf*s25, r*Pf*MSP8, or r*Pf*s25/8 antigens. Additional sets were stimulated with Concanavalin A (Con A) or left unstimulated to serve as positive and negative controls, respectively. Proliferative responses of antigen-specific T cells were determined using a standard [$^3$H]-thymidine incorporation assay.

As shown in Fig 1(A)–1(D), T cells from all groups immunized with any formulation containing r*Pf*MSP8 (r*Pf*s25 + *Pf*MSP8 admixture, r*Pf*s25/8, r*Pf*MSP8) demonstrated similar and strong proliferative responses when stimulated with r*Pf*MSP8; all were significantly higher than those observed by cells from corresponding adjuvant control mice. This was true irrespective of antigen dose or adjuvant. Similarly, T cells from groups immunized with *Pf*s25-containing formulations (r*Pf*s25 alone, r*Pf*s25 + *Pf*MSP8 admixture, r*Pf*s25/8) demonstrated similar and specific proliferative responses when stimulated with r*Pf*s25 that were significantly higher than corresponding control mice. Again, this was true irrespective of antigen dose or adjuvant, with one exception. Proliferation of T cells from mice immunized with 2.5 μg of r*Pf*s25/8 formulated with GLA-SE and stimulated in vitro with r*Pf*s25 was low and not significantly different relative to adjuvant control mice (Fig 1C). This could potentially be due to a shift of the T cell response toward epitopes in r*Pf*MSP8 at the higher dose. With Alum-based formulations, the slightly higher than expected proliferation of cells from r*Pf*s25-vaccinated mice when stimulated with r*Pf*MSP8 and vice versa (Fig 1A and 1B) may be due to a shared epitope(s) between the two antigens associated with a common leader and linker sequence. As expected, cells harvested from all antigen-immunized mice demonstrated high proliferative responses when stimulated with chimeric r*Pf*s25/8 (S1 Fig). These were generally additive of the domain-specific responses. Collectively, these data demonstrate that T cells from CB6F1/J immunized mice recognize epitopes present in both *Pf*s25 and *Pf*MSP8 domains.

### Antigen-specific T cells induced by Alum- and GLA-SE-based formulations are skewed toward Th$_2$ and Th$_1$ profiles, respectively, in a *Pf*MSP8-dependent manner

The effect of adjuvant on the type of antigen-specific T helper cells (Th$_1$ vs. Th$_2$) induced by immunization was evaluated based on cytokine production following antigen re-exposure. Similar to the T cell proliferation studies, splenocytes were collected and stimulated *in vitro* with r*Pf*s25, r*Pf*s25/8, r*Pf*MSP8 or cultured in media alone. Stimulation with Con A served as a positive control. Culture supernatants were collected and analyzed for production of IL-5, TNFα, IFNγ, IL-2 and IL-4 via a multiplex assay (Luminex). As expected, stimulation with Con A elicited detectible and similar responses in cells from all groups (S1 Table). As shown in Fig 2, cells from most antigen-immunized groups secreted detectible levels of IL-5, IFNγ and TNFα when stimulated with r*Pf*s25/8. However, there was a clear effect of adjuvant on the relative level of each of these cytokines.

Irrespective of antigen dose, T cells from groups immunized with Alum-based formulations produced primarily IL-5 (Fig 2A). These levels were similar for all formulations containing

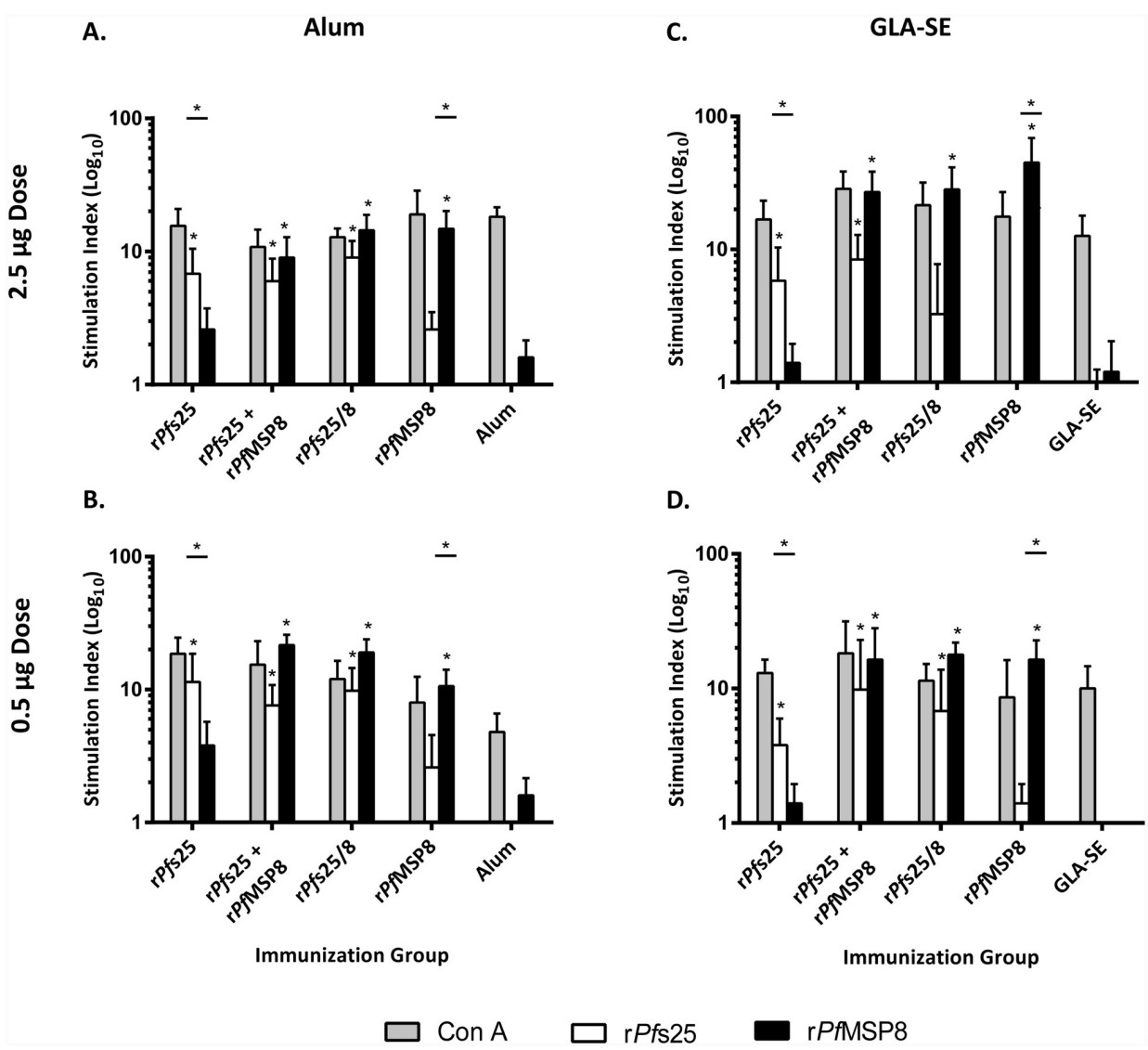

**Fig 1. Both rPfs25 and rPfMSP8 domains elicit antigen-specific T cell responses irrespective of vaccine formulation.** Splenocytes ($2 \times 10^5$/ well) harvested from groups of CB6F1/J mice immunized as indicated were stimulated *ex vivo* in triplicate with r*Pf*MSP8 (2 μg/ well) or r*Pf*s25 (2 μg/ well) for 96 hrs. Additional sets were stimulated with Con A (0.2 μg/well) or incubated in normal media as positive and negative controls, respectively. [$^3$H]-thymidine (1 μCi/ well) was added for the final 18 hours. Average counts of incorporated [$^3$H]-thymidine were measured for each stimulation condition and converted into a Stimulation Index (SI) that represents the fold change in proliferation of the indicated condition over the corresponding control wells (media alone). Graphs depict the mean SI +/- standard deviation. Asterisks directly over a single bar represent significant differences between the indicated groups relative to the corresponding adjuvant control group (Kruskal-Wallis Test; $P < 0.05$ considered significant). Asterisks over horizontal lines indicate significant differences between the two groups (Mann-Whitney $U$ Test; $P < 0.05$ considered significant).

*Pf*MSP8 and significantly higher compared to levels produced by T cells from Alum control mice. Of interest, cells from groups immunized with r*Pf*s25 formulated in Alum produced only low levels of IL-5, not significantly different than adjuvant control mice.

 The response of T cells from mice immunized with vaccines formulated with GLA-SE was robust at both antigen doses and marked by a more diversified cytokine profile (Fig 2). Cells from mice immunized with *Pf*MSP8-containing vaccines with GLA-SE as adjuvant produced IL-5 at significantly higher levels relative to the control group (Fig 2A). Of note, cells from

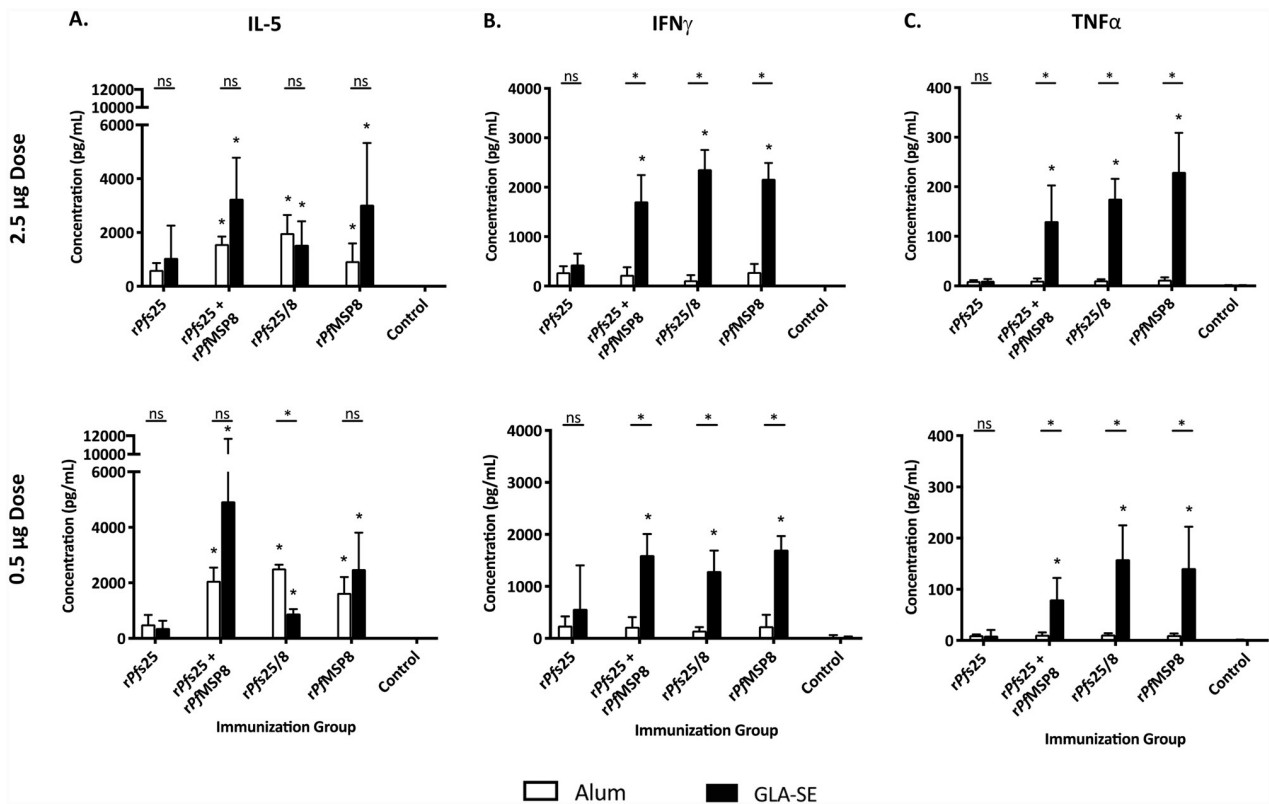

**Fig 2. Cytokine production by CD4⁺ T cells reveals different T helper profiles as a function of adjuvant in a *Pf*MSP8-dependent manner.**
Splenocytes (5 x 10⁵/well) were harvested from groups of CB6F1/J mice immunized as indicated and stimulated *ex vivo* in the presence of r*Pf*s25/8 (2 μg/well) or in media alone for 96 hours. Culture supernatants were collected and analyzed for production of **A)** IL-5, **B)** IFNγ and **C)** TNFα using a multiplex assay (Luminex®). To calculate the final concentration of each analyte, levels found in the corresponding unstimulated conditions were subtracted as background. Graphs depict the mean concentration of each analyte +/- standard deviation. Asterisks directly over single bars represent significant differences between the indicated groups compared to the corresponding adjuvant control group (Kruskal-Wallis Test; $P < 0.05$ considered significant). Asterisks over horizontal lines signify significant differences between the two indicated groups (Mann-Whitney *U* test; $P < 0.05$ considered significant; ns, not significant).

mice immunized with unfused r*Pf*s25 formulated with GLA-SE produced low levels of IL-5 that were not statistically above background controls. In pairwise comparisons, the IL-5 levels elicited by GLA-SE-based formulations were similar to those produced by the corresponding Alum-based formulations at both doses. In contrast, groups immunized with *Pf*MSP8-containing vaccines formulated with GLA-SE also produced both IFNγ (Fig 2B) and TNFα (Fig 2C) at levels significantly higher than the corresponding Alum-based antigen groups and GLA-SE controls. This was not true for cells from mice immunized with unfused r*Pf*s25 in GLA-SE, which did not produce significant quantities of either IFNγ or TNFα.

The production of IL-2 and IL-4 by antigen-specific T cells was also assessed following stimulation with r*Pf*s25/8. Production of both analytes was low in all vaccine formulations, with no significant differences observed as a function of immunizing antigen, dose and/or adjuvant (S2 Table). The observation that cytokine production is adjuvant and carrier-dependent was further confirmed by results of domain-specific stimulation with r*Pf*MSP8 alone (S2 Fig) which yielded results similar to stimulation with r*Pf*s25/8. Therefore, in some contrast to the proliferative responses observed upon stimulation with both r*Pf*s25 and r*Pf*MSP8 domains, cytokine production was driven primarily by epitopes present in *Pf*MSP8 domain with choice of adjuvant influencing the profile.

## Fusion of r*Pf*s25 to the *Pf*MSP8 carrier elicits strong anti-*Pf*s25 antibody responses and alleviates antigenic competition

It is well established that transmission-reducing activity of *Pf*s25-based vaccines is primarily antibody mediated. As such, the magnitude and specificity of the antibody responses induced by each vaccine formulation over time was evaluated. Sera were collected following each of three s.c. immunizations and domain-specific IgG titers determined. As depicted in Fig 3A, immunization with 2.5 μg dose of r*Pf*s25, r*Pf*s25 + r*Pf*MSP8, or r*Pf*s25/8 adjuvanted with Alum elicited strong and comparable anti-*Pf*s25 IgG titers that were significantly boosted over time. At the 0.5 μg dose in Alum (Fig 3B), high anti-*Pf*s25 IgG were also generated in response to immunization with r*Pf*s25. However, the anti-*Pf*s25 IgG response was impaired in mice

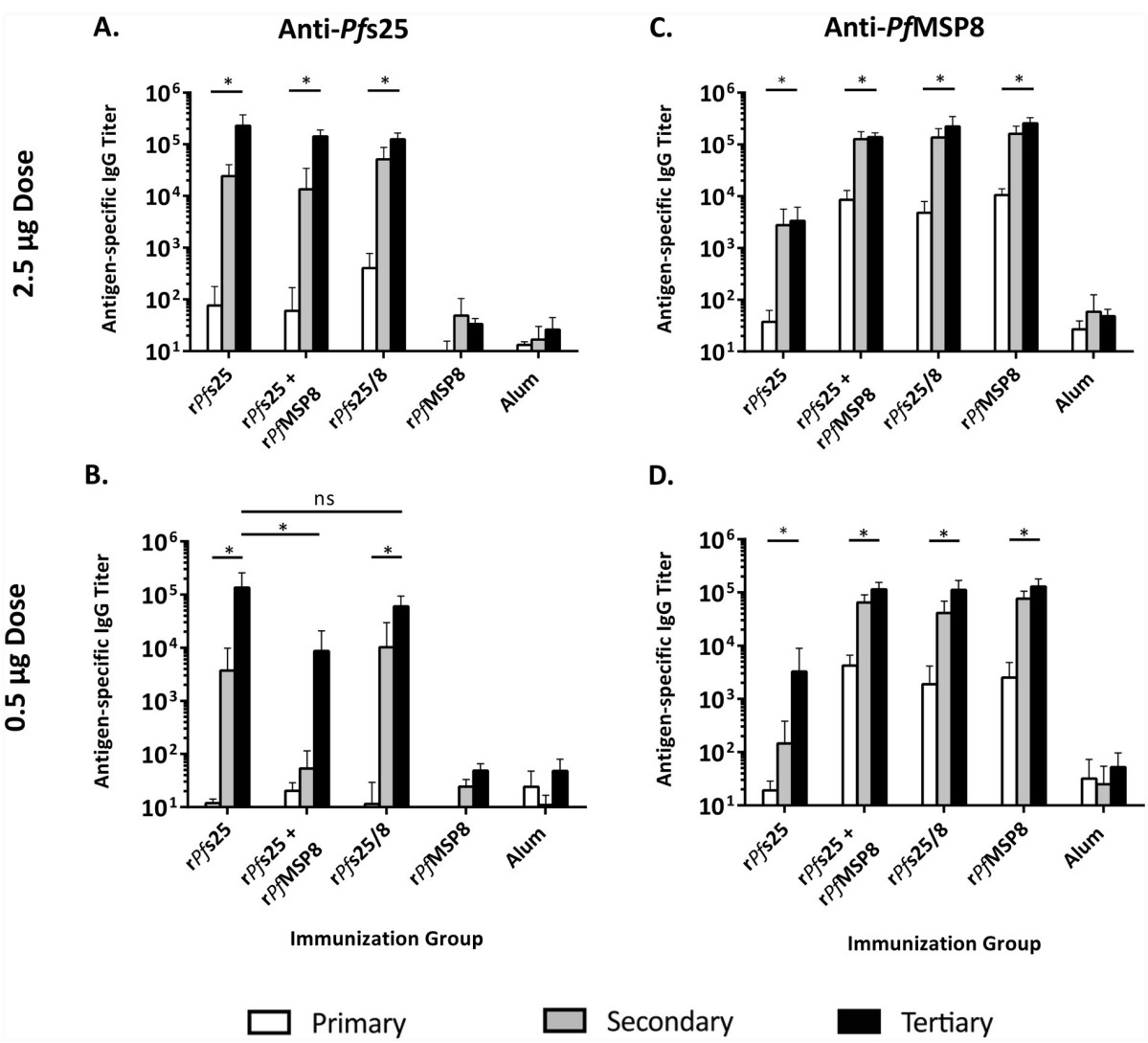

**Fig 3. Antigen-specific IgG titers elicited by Alum-based formulations.** CB6F1/J sera collected 3 weeks following each subcutaneous immunization were analyzed for antigen-specific antibodies by ELISA using plates coated with r*Pf*s25 (**A and B**) or *Pf*MSP8 (**C and D**) (0.25 μg/well). Graphs depict the mean IgG titers +/- standard deviation. Asterisks over horizontal lines within an immunization group indicate significant boosting of antigen-specific IgG titers over time (Friedman Test; *P* < 0.05 considered significant). Asterisks over horizontal lines comparing different immunization groups indicate significant differences between final titers achieved by those groups (Kruskal Wallis Test, *P* < 0.05 considered significant; ns, not significant).

immunized with the admixture of r*Pf*s25 + r*Pf*MSP8 relative to animals immunized with r*Pf*s25 alone, indicative of competition between r*Pf*s25 and r*Pf*MSP8 when administered as separate, non-fused components. Importantly, this competition was eliminated upon immunization with 0.5 μg of the chimeric r*Pf*s25/8 in Alum with a strong anti-*Pf*s25 IgG response comparable to that observed in the r*Pf*s25 group. Immunization with either dose of r*Pf*s25/8, the r*Pf*s25 + r*Pf*MSP8 admixture or *Pf*MSP8 antigens formulated in Alum induced strong and similar titers against the highly immunogenic r*Pf*MSP8 carrier, which were increased significantly over time (Fig 3C and 3D). Of note, antibodies induced by immunization with r*Pf*s25 alone exhibited some reactivity with r*Pf*MSP8 (Fig 3C and 3D). This reactivity is associated with a shared epitope(s) present within the His-tag and linker that are common to both antigens. This reactivity is relatively low, representing only 1–2% of the overall anti-*Pf*s25 titer induced by immunization with unfused r*Pf*s25 (Fig 3A and 3B).

Likewise, the anti-*Pf*s25 response induced by immunization with 2.5 μg of r*Pf*s25, r*Pf*s25 + r*Pf*MSP8 admixture or r*Pf*s25/8 formulated in GLA-SE was assessed (Fig 4A). Responses were detected in all groups that were significantly boosted over time. However, there was a 10-fold reduction in final anti-*Pf*s25 titer in mice immunized with the r*Pf*s25 + r*Pf*MSP8 admixture relative to the group immunized with r*Pf*s25 alone, highlighting competition between antigens. Importantly, this response was restored in mice immunized with the chimeric r*Pf*s25/8, resulting in a higher final titer relative to the r*Pf*s25 immunized mice group. Antibody responses elicited by immunization with these antigens formulated at 0.5 μg also increased significantly over time (Fig 4B). However, the anti-*Pf*s25 response elicited by r*Pf*s25 at the 0.5 μg dose was more than 10-fold lower than that induced at the 2.5 μg dose. Again, anti-*Pf*s25 titers were further decreased in the admixture group. Similar to the 2.5 μg dose group, immunization with chimeric r*Pf*s25/8 was able restore this response with final anti-*Pf*s25 titers even greater than those observed in mice immunized with unfused r*Pf*s25. Despite divergent responses against *Pf*s25, the anti-*Pf*MSP8 responses elicited in mice immunized with r*Pf*s25 + r*Pf*MSP8, r*Pf*s25/8 or r*Pf*MSP8 carrier alone formulated with GLA-SE were strong and comparable irrespective of dose and was significantly boosted over time (Fig 4C and 4D). Antibody titers measured against r*Pf*s25/8 coated wells were generally additive of the two individual domain-specific responses (S3 Fig).

Finally, the effectiveness of Alum vs GLA-SE as adjuvant in mice immunized with r*Pf*s25 or the chimeric r*Pf*s25/8 was considered. As shown in Fig 5, both r*Pf*25-containing vaccines induced high and comparable titers against *Pf*s25 when formulated with Alum irrespective of dose. However, there were significant differences in final titer induced by the two antigens when GLA-SE was used as an adjuvant. Here, the final anti-*Pf*s25 titer induced in the r*Pf*s25-immunized group was 10-fold lower than the corresponding group formulated with Alum. Importantly, chimeric r*Pf*s25/8 formulated with GLA-SE elicited significantly higher final titers of anti-*Pf*s25 IgG relative to immunization with unfused r*Pf*s25 at the same dose; these titers were comparable to those induced by Alum-based formulations. Together, these data indicate that Alum was an equally potent adjuvant for both vaccine antigens, while maximal anti-*Pf*s25 responses elicited by GLA-SE based formulations depended on genetic fusion of r*Pf*s25 to the r*Pf*MSP8 carrier protein.

## Switch from Alum to GLA-SE as adjuvant for *Pf*s25-based vaccines shifts the B cell response to the production of cytophilic IgG in a *Pf*MSP8-dependent manner

In addition to titer, the functionality of vaccine-induced IgG may be influenced by heavy chain subclass depending on adjuvant selection. The profile of IgG subclasses in the final sera

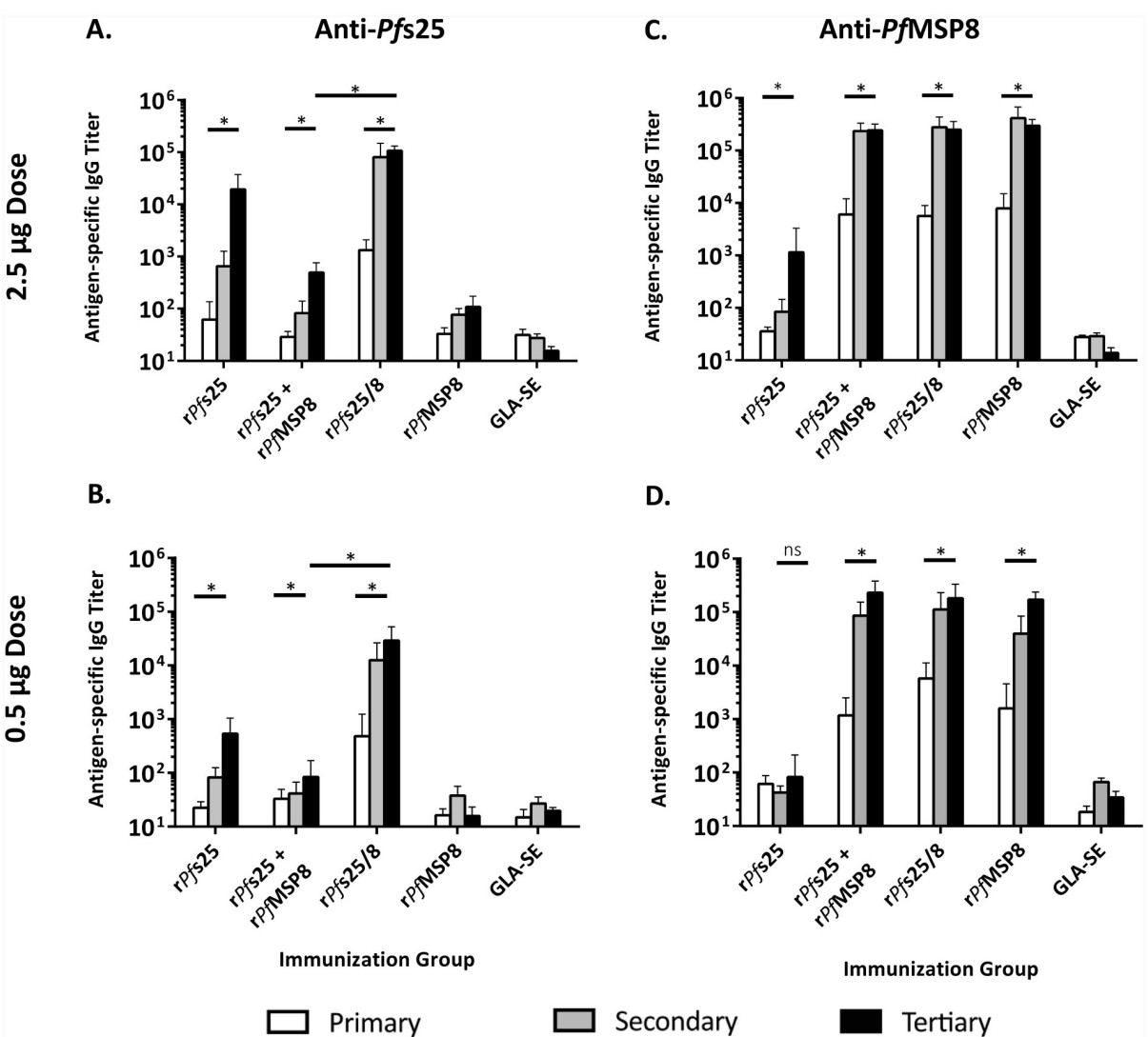

**Fig 4. Antigen-specific IgG titers elicited by GLA-SE-based formulations.** CB6F1/J sera collected 3 weeks following each subcutaneous immunization were analyzed for antigen-specific antibodies by ELISA using plates coated with r*Pf*s25 **(A and B)** or *Pf*MSP8 **(C and D)** (0.25 µg/well). Graphs depict the mean IgG titers +/- standard deviation. Asterisks over horizontal lines within an immunization group indicate significant boosting of antigen-specific IgG titers over time (Friedman Test; $P < 0.05$ considered significant). Asterisks over horizontal lines comparing different immunization groups indicate significant differences between final titers achieved by those groups (Kruskal Wallis Test, $P < 0.05$ considered significant).

from each vaccine group was measured by ELISA using plates coated with r*Pf*s25/8 and secondary antibodies specific for IgG1, IgG2a/c, IgG2b and IgG3. As shown in Fig 6, vaccines formulated with Alum, regardless of antigen or dose, elicited antibodies primarily of the IgG1 subclass, with low but detectible IgG2a/c, IgG2b and IgG3. In the same way, immunization with r*Pf*MSP8-containing vaccines formulated with GLA-SE, irrespective of dose, produced high and similar levels of IgG1 compared to the Alum-formulated counterparts. One exception was noted in the r*Pf*s25 + *Pf*MSP8 admixture group that showed a modest but statistically significant reduction in IgG1. In stark contrast to the IgG profiles induced by Alum-based formulations, mice immunized with GLA-SE-based formulations also produced significantly higher

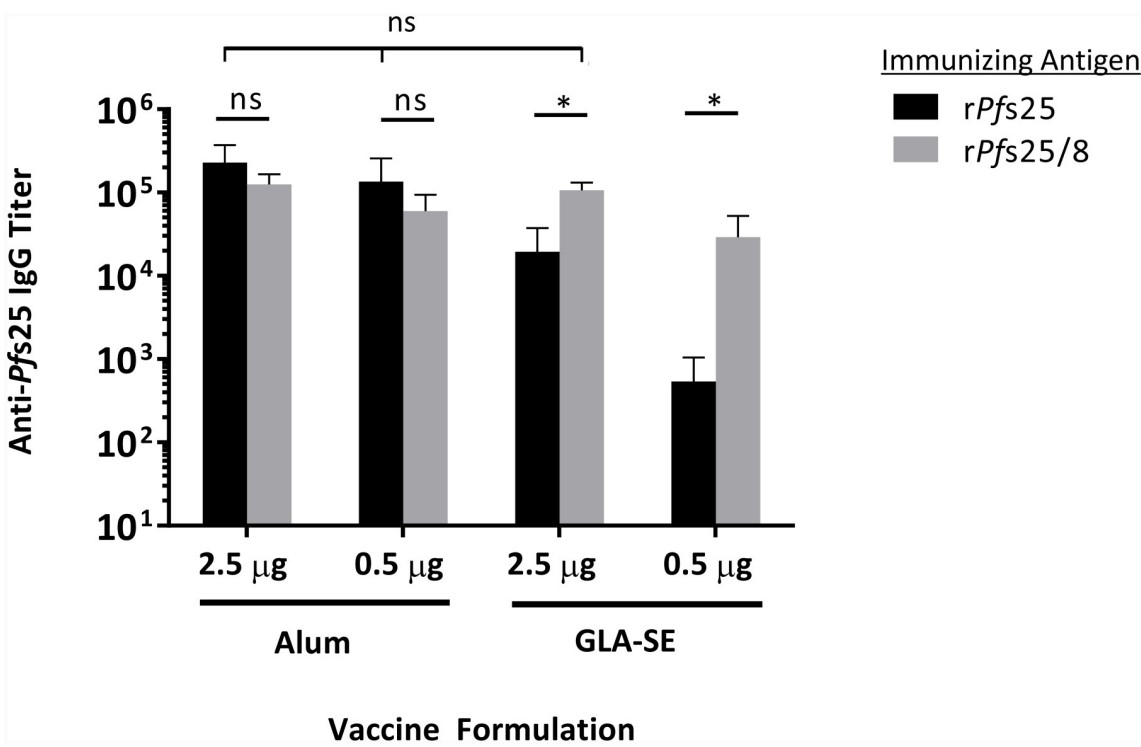

**Fig 5. Final anti-*Pf*s25 IgG titers induced by immunization with r*Pf*s25 and r*Pf*s25/8 as a function of dose and adjuvant.** Direct comparison of final anti-*Pf*s25 IgG titers induced by r*Pf*s25 vs chimeric r*Pf*s25/8 antigens when formulated as indicated. Graphs depict the mean IgG titers +/- standard deviation. Comparisons of two groups differing only by antigen identity were conducted and significant differences are indicated by asterisks over horizontal lines (Mann-Whitney *U* Test; $P < 0.05$ considered significant). Comparisons of three or more groups were conducted as indicated and no significant differences were found (Kruskal Wallis Test, $P < 0.05$ considered significant; ns, not significant).

levels of antigen-specific IgG2a/c, IgG2b and IgG3 relative to the Alum-formulated counterparts. This was not true for r*Pf*s25-immunized animals, which primarily produced antigen-specific IgG1 when formulated with either Alum or GLA-SE. These results demonstrate a strong influence of adjuvant on IgG subtype profile generated in response to vaccination, with IgG class-switching in the GLA-SE formulations dependent on the presence of the r*Pf*MSP8 carrier.

### Transmission-reducing activity (TRA) of IgG induced by immunization with r*Pf*s25-based vaccines formulated with Alum vs GLA-SE

In addition to the magnitude and profile of antibody responses, demonstrating the ability of *Pf*s25-containing vaccines to induce IgG that inhibits development of sexual-stage parasites in the mosquito vector is important. To test the relative functionality of IgG induced by r*Pf*s25 and r*Pf*s25/8 vaccines, total IgG was purified from pools of sera derived from each vaccine group and tested in the SMFA at a concentration of 750 μg/ml. As depicted in Table 1, all Alum-based formulations elicited potent and comparable TRA relative to the IgG derived from adjuvant-immunized control groups. Similarly, IgG from r*Pf*s25/8 + GLA-SE immunized groups also demonstrated potent TRA relative to control IgG. As predicted based on analysis of anti-*Pf*s25 antibody responses, groups immunized with r*Pf*s25 formulated with GLA-SE had much lower TRA that were not statistically different from controls.

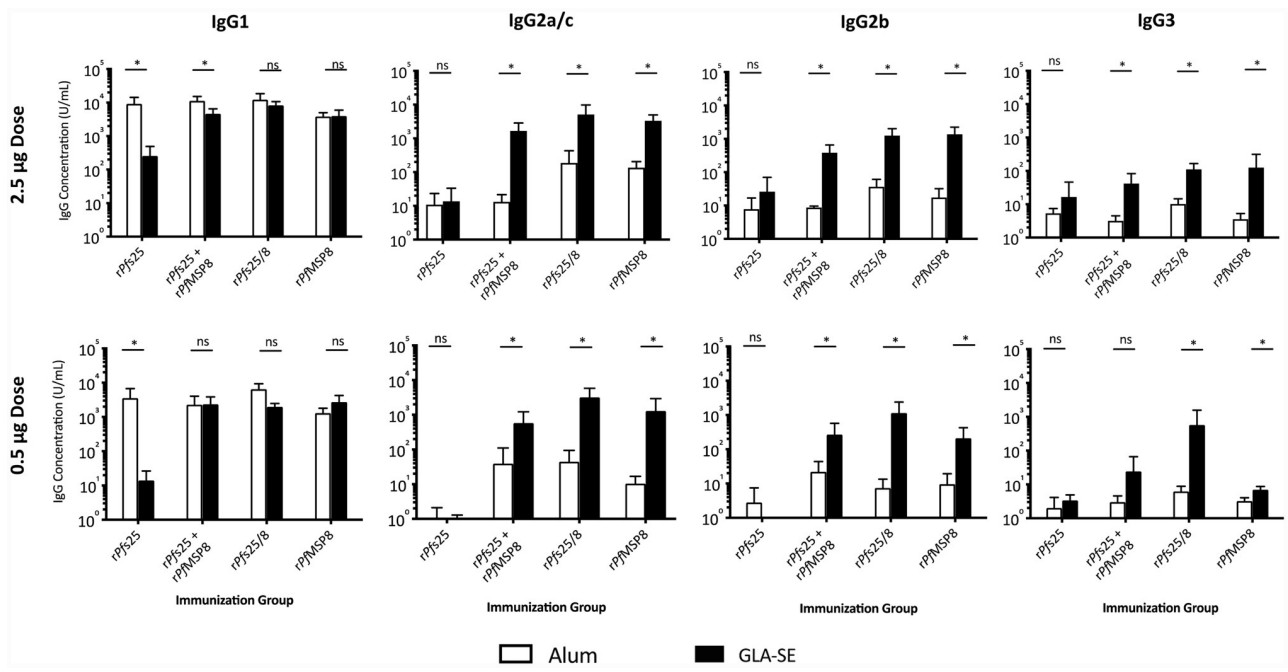

**Fig 6. Profile of IgG subclasses of vaccine-induced IgG is dependent upon adjuvant and antigen.** CB6F1/J sera harvested four weeks following the final immunization were analyzed for relative levels of the indicated IgG subclasses by ELISA using plates coated with r*Pf*s25/8 (0.25 µg/well). Graphs depict the mean IgG subclass concentration +/- standard deviation. Comparisons of two groups that differed by adjuvant only were conducted and asterisks over horizontal lines indicate significant differences between groups (Mann-Whitney *U* Test; *P* < 0.05 considered significant; ns, not significant).

**Bivalent formulations containing r*Pf*s25/8 and r*Pf*MSP1/8 elicit strong antibody titers against both fusion partners in outbred mice, with no indication of antigen competition.**

Based on the above immunogenicity and functionality data, chimeric r*Pf*s25/8 was evaluated in combination with r*Pf*MSP1/8, which elicits potent merozoite invasion inhibitory antibodies. The immunogenicity of r*Pf*s25/8 + r*Pf*MSP1/8 (2.5 µg each antigen/dose) was compared to

**Table 1. Transmission-reducing activity of vaccine-induced IgG is a function of antigen and adjuvant.**

| Immunization Group | | | IgG level (µg/ml) | Transmission-reducing activity | | | |
|---|---|---|---|---|---|---|---|
| Adjuvant | Antigen | Dose | | % inhibition | 95% CI (low) | 95% CI (high) | *P* value (vs. control pool) |
| **Adjuvant Control Sera** | N/A | N/A | 750 | 0 | | | |
| **Alum** | r*Pf*s25 | 2.5 µg | 750 | 100.0 | 98.7 | 100.0 | 0.001 |
| | r*Pf*s25/8 | 2.5 µg | 750 | 99.3 | 97.5 | 100.0 | 0.001 |
| | r*Pf*s25 | 0.5 µg | 750 | 99.3 | 97.6 | 100.0 | 0.001 |
| | r*Pf*s25/8 | 0.5 µg | 750 | 98.0 | 95.4 | 99.5 | 0.001 |
| **GLA-SE** | r*Pf*s25 | 2.5 µg | 750 | 46.5 | -23.1 | 75.5 | 0.128 |
| | r*Pf*s25/8 | 2.5 µg | 750 | 97.4 | 90.1 | 100.0 | 0.001 |
| | r*Pf*s25 | 0.5 µg | 750 | 46.5 | -25.8 | 79.5 | 0.135 |
| | r*Pf*s25/8 | 0.5 µg | 750 | 89.4 | 75.4 | 95.9 | 0.001 |

Total IgG from pools of tertiary sera collected from CB6F1/J mice immunized with r*Pf*s25 or r*Pf*s25/8 formulated as indicated was purified and tested in a standard membrane feeding assay at a concentration of 750 µg/ml. Each IgG sample was fed to 20 mosquitos. The best estimates and 95% confidence intervals (CIs) of percent inhibitions and *P* values were calculated for each pool compared to IgG derived from adjuvant-immunized control mice.

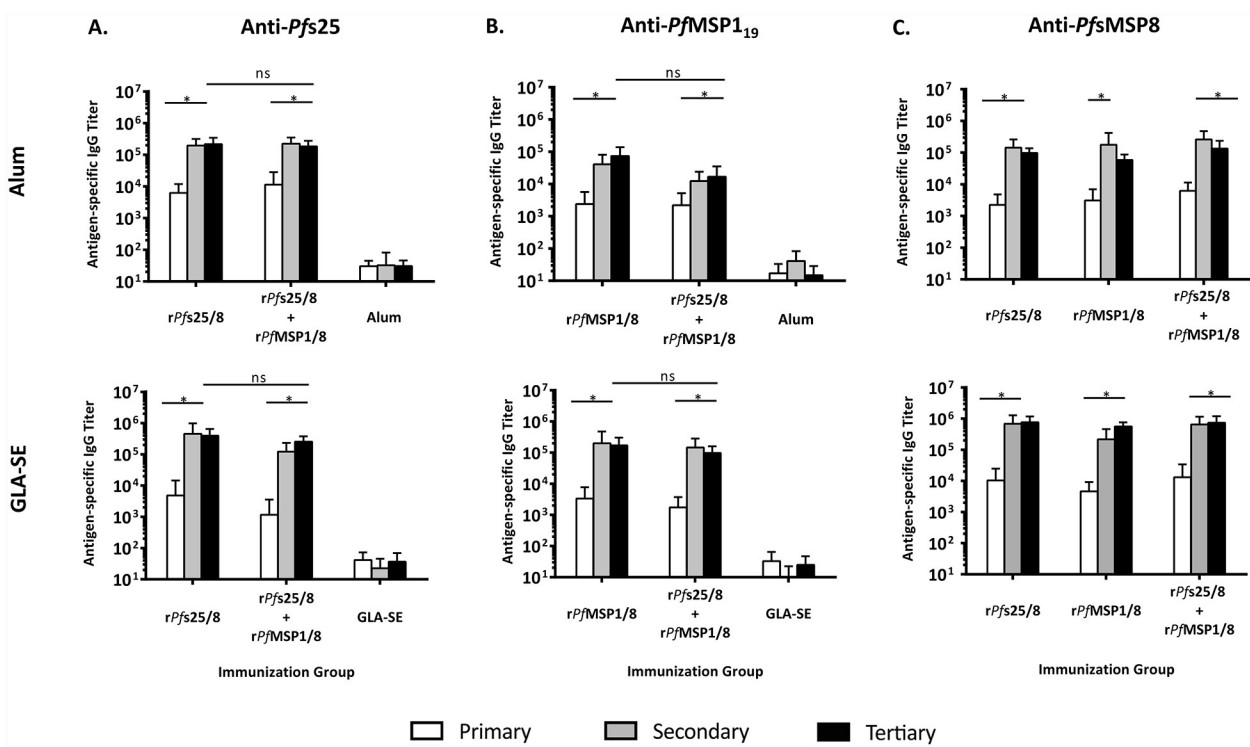

**Fig 7. Bivalent formulations containing rPfs25/8 and rPfMSP1/8, irrespective of adjuvant, elicit strong titers against both fusion partners with no indication of antigen competition relative to corresponding monovalent vaccines.** Sera harvested from groups of CD1 mice (5 male, 5 female) immunized as indicated were analyzed for antigen-specific IgG by ELISA using plates coated with **A)** rPfs25, **B)** rGST-PfMSP1$_{19}$ or **C)** rPfMSP8 (0.25 μg/well). Sera from each adjuvant control group were measured against the corresponding chimeric antigen (rPfs25/8 or rPfMSP1/8). Graphs depict mean IgG titers +/- standard deviation. Asterisks over horizontal lines within an immunization group indicate significant boosting of antigen-specific responses over time (Friedman Test; $P < 0.05$ considered significant). Final IgG titers induced against each component in the bivalent formulations were compared to the corresponding monovalent group (Mann-Whitney $U$ test; $P < 0.05$ considered significant; ns, not significant).

corresponding monovalent vaccines (2.5 μg/dose) when adjuvanted with Alum or GLA-SE. Immunizations were conducted in male (n = 5) and female (n = 5) outbred (CD1) mice in order to i) assess the consistency of vaccine-induced responses in a genetically diverse population, ii) evaluate sex as a variable with potential to influence immune responses, and iii) determine the impact of concurrent immunization with two subunit vaccines fused to the same rPfMSP8 carrier. Mice were immunized three times and sera collected following each immunization. These samples were then analyzed for antigen-specific titers against both *Pf*s25 and *Pf*MSP1$_{19}$ fusion partners, well as the *Pf*MSP8 carrier.

As shown in Fig 7A–7C, all antigen-immunized groups mounted strong antigen-specific antibody response against component domains contained in the formulation (*Pf*MSP1$_{19}$, *Pf*s25, and/ or *Pf*MSP8). These antigen-specific antibody responses were significantly boosted over time in all groups. Importantly, final domain-specific IgG titers elicited by immunization with the bivalent formulations were comparable to those induced by the corresponding monovalent vaccines, regardless of adjuvant or antigen (Fig 7). Following three immunizations, the anti-*Pf*MSP1$_{19}$ and anti-*Pf*s25 titers in male and female mice in each immunization group were comparable, indicating that host sex did not influence vaccine immunogenicity (S4 Fig). Overall, these data indicate that concurrent responses against B cell determinants within two

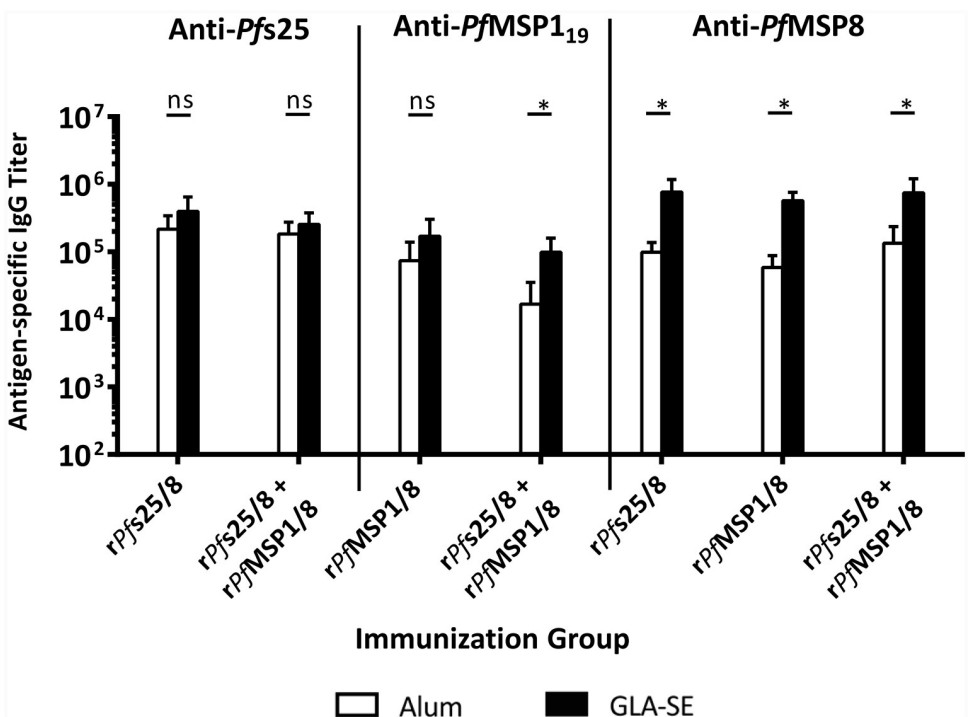

**Fig 8. Final antigen-specific IgG titers elicited in monovalent and bivalent vaccine formulations as a function of adjuvant.** Titers of antigen-specific IgG following the final subcutaneous immunization were measured for each group via ELISA using plates coated with r*Pf*s25 or rGST-*Pf*MSP1$_{19}$ (0.25 μg/well). Graphs depict mean IgG titers +/- standard deviation. To assess the effect of adjuvant on the IgG titer, direct comparisons were made between corresponding antigen groups formulated with either Alum or GLA-SE. Asterisks over horizontal lines indicate significant differences between indicated groups (Mann-Whitney *U* test; $P < 0.05$ considered significant; ns, not significant).

different *Pf*MSP8 fusion partners can be effectively induced by a bivalent vaccine without antigenic competition.

The effect of adjuvant on the domain-specific IgG titers induced by each antigen formulation was compared. As shown in Fig 8, the magnitude of the anti-*Pf*s25 response was high and comparable in monovalent vs. bivalent vaccines, and similar between Alum- and GLA-SE-based formulations. These findings indicate that immunization with r*Pf*s25/8 induces strong anti-*Pf*s25 responses that are independent of adjuvant selection and are not inhibited by the presence of r*Pf*MSP1/8 in the formulation. In contrast, anti-*Pf*MSP1$_{19}$ titers induced by the bivalent vaccine were significantly higher when formulated with GLA-SE vs. Alum. In addition, responses to the *Pf*MSP8 carrier were significantly higher when formulated with GLA-SE vs. Alum across all immunization groups. As such, the strength of anti-*Pf*MSP1$_{19}$ titers generated by immunization with r*Pf*MSP1/8 was dependent upon adjuvant, with GLA-SE-based formulations inducing superior responses.

As above, the profile of anti-*Pf*MSP8 IgG induced by each formulation was evaluated to determine the relative levels of IgG1, IgG2a/c, IgG2b and IgG3 subtypes. As shown in Fig 9, strong and similar levels of IgG1 were detected in all groups regardless of antigen formulation or adjuvant. All Alum-based formulations induced detectible but very low levels of IgG2a/c and IgG2b, consistent with the expected Th$_2$ associated response. In contrast, strong and significant production of IgG2a/c, IgG2b and IgG3 was observed in all GLA-SE-based

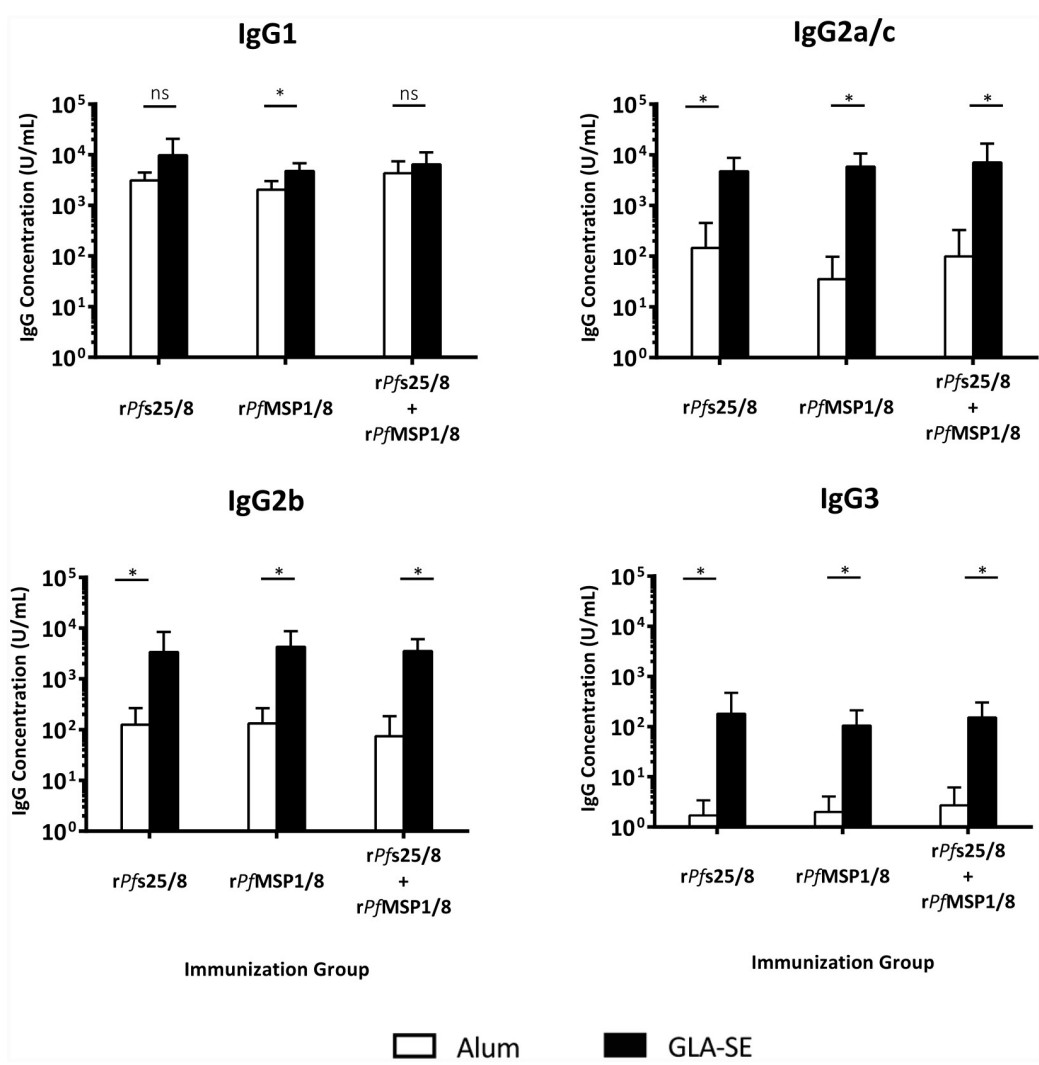

**Fig 9. Profile of IgG subclasses of vaccine-induced IgG is dependent upon adjuvant and antigen.** Sera harvested four weeks following the final immunization were analyzed for relative levels of the indicated subclasses of IgG via ELISA using plates coated with r*Pf*MSP8 (0.25 µg/well). Graphs depict mean IgG subclass concentration (U/ml) +/- standard deviation. Comparisons of two corresponding groups that differed by adjuvant only were conducted and asterisks over horizontal lines indicate significant differences between groups (Mann-Whitney *U* Test; *P* < 0.05 considered significant).

formulations. These data confirm results from studies in inbred mice with vaccines utilizing *Pf*MSP8 as a carrier (Fig 6), showing that the production of cytophilic IgG occurs in the context of the Th$_1$-baising adjuvant, GLA-SE, and leads to a more diverse IgG subclass profile.

## Discussion

Though conceptually simple, the rational production of a multivalent, multistage subunit malaria vaccine is a challenge. The first requirement is that recombinant forms of each component be produced so that they bear proper conformation and induce antibodies that demonstrate functionality against the targeted stage(s) of plasmodial parasites. The production of recombinant antigens with routine heterologous expressions systems has often resulted in low yields and/or inconsistent, misfolded products incapable of producing protective antibodies to

neutralizing, conformational B cell epitopes. [43–46]. This is true for *Pf*s25, but transmission blocking antibodies have been induced by r*Pf*s25 produced in scalable yeast [43, 47, 48] and baculovirus [49] systems and recently with a non-glycosylated, folded product of an *E. coli* expression system [50].

An equally important impediment to overcome is the inherently poor immunogenicity of many malaria vaccine candidates. This is often attributed to the small size of the antigen or, more often, the inability to elicit effective CD4$^+$ T cell responses capable of providing adequate help to B cells for the production of protective antibodies. In some cases, this deficit may be due to a lack of immunogenic CD4$^+$ T cell epitopes. However, it may also be due to the presence of complex tertiary structures that inhibit antigen processing and presentation, such as highly constrained EGF-like domains. This appears to be the case for *Pf*MSP1$_{19}$ [51, 52]. Further complicating the task is the need to induce protective and durable responses against each component that can be maintained when formulated in a multiantigen combination. Antigenic competition between co-administered components has been noted in several studies [4–6]. In the recent Phase III RTS,S trial, one of several contributing factors to lower responses in children compared to adults may relate to the administration of RTS,S concurrently with other childhood vaccines through the Expanded Program for Immunization (EPI) [53, 54].

Common strategies to address issues relating to immunogenicity, include i) use of heterologous carrier proteins, ii) formulation with potent adjuvants, and iii) optimization of vaccine dose, route of administration and/ or timing of vaccinations. These considerations, as well as the nature of the target itself, are likely to impact overall immunogenicity of a given formulation. Variations of these parameters were evaluated in an effort to improve the efficacy and duration of responses generated by immunization with the *Pf*s25-Exoprotein A conjugate [55], the leading *Pf*s25-based clinical candidate. Results of those studies indicated that anti-*Pf*s25 responses were influenced by both adjuvant and specific carrier selected [55]. Similarly, our comparative immunogenicity studies indicate that the induction of potent anti-*Pf*s25 responses is significantly influenced by the presence of the *Pf*MSP8 carrier as well as by adjuvant formulation.

We began a systematic evaluation of our candidate antigens by assessing how these parameters affected cellular responses. Using a [$^3$H]-thymidine incorporation assay to determine the specificity of vaccine-induced cellular responses, we showed that immunization with any of the *Pf*MSP8-containing vaccines induced significant and similar levels of antigen-specific T cell proliferation against *Pf*MSP8. This was expected, as previous studies conducted in both inbred and outbred mice demonstrated the presence of potent CD4$^+$ T cell epitopes within the *Pf*MSP8 carrier [24, 25]. Interestingly, proliferative responses against *Pf*s25 were also detected following immunization with all three r*Pf*s25-containing vaccines, though at a lower magnitude relative to *Pf*MSP8. Nevertheless, this indicated that *Pf*s25 possesses one or more MHC II epitopes capable of inducing *Pf*s25-specific CD4$^+$ T cell responses in mice, even in the absence of a carrier. Indeed, *Pf*s25 contains at least one epitope predicted to bind MHC II. Unlike *Pf*MSP1$_{19}$, the CD4$^+$ epitope(s) within *Pf*s25 appear(s) to be available for processing and presentation despite the highly constrained nature of this antigen. The specificity of CD4$^+$ T cells induced by these formulations was not affected by adjuvant selection.

The phenotype of T cells generated in response to *Pf*s25-based vaccines was influenced by adjuvant. Alum-based formulations containing r*Pf*s25/8, regardless of dose, elicited significant levels of Th$_2$-associated cytokine, IL-5, and very low levels of Th$_1$-associated cytokines, IFNγ and TNFα. This was expected, as Alum is a known Th$_2$-biasing adjuvant. Downstream analysis of the IgG subclasses induced by vaccines adjuvanted with Alum reflected this Th$_2$-biasing effect, as the vast majority of antigen-specific antibodies were IgG1. Vaccines formulated with GLA-SE, an adjuvant known to drive responses to a Th$_1$ phenotype, elicited T cells that

produced significantly elevated levels of Th$_1$-associated IFNγ and TNFα in comparison to Alum-based formulations, in a *Pf*MSP8-dependent manner. These T cells also produced IL-5 at similar levels to those achieved by Alum-based formulations. However, the elevated levels of Th$_1$-associated cytokines in GLA-SE based formulations influenced downstream class switching, leading to a more diversified IgG profile of antigen-specific antibodies that featured significant increases in cytophilic IgG2a/c, as well as IgG2b and IgG3.

We assessed the domain-specific responses of total IgG induced by the various vaccine formulations as an indicator of potential efficacy. All formulations containing r*Pf*MSP8 resulted in high and comparable titers of anti-*Pf*MSP8 IgG irrespective of adjuvant or dose. However, the induction of optimal anti-*Pf*s25 humoral immunity was dependent on both carrier and adjuvant. Consistent with the proliferation data, immunization with r*Pf*s25 elicited high anti-*Pf*s25 antibody titers when formulated with Alum at both antigen doses. In contrast, only modest titers of anti-*Pf*s25 IgG were elicited by unfused r*Pf*s25 when formulated with GLA-SE, despite the detection of proliferative T cell responses in these groups. These results differ from a previous study in which *Chlamydomonas reinhardtii*-produced r*Pf*s25 formulated with GLA-SE effectively induced anti-*Pf*s25 antibodies that exhibited transmission-reducing activity in the SMFA [56]. However, this discrepancy may be a result of significant differences in the total amount of r*Pf*s25 administered in the two studies (50 μg vs 7.5 μg). In three of the four groups immunized with the admixture of r*Pf*s25 and r*Pf*MSP8, we observed a notable reduction in anti-*Pf*s25 titers. These data indicate that antigenic competition is a potential problem and responses to r*Pf*s25 may be impaired by the presence of additional immunogenic vaccine components.

In agreement with our previous r*Pf*MSP1/8 studies, immunization with r*Pf*s25/8 elicited strong humoral responses against both the carrier and *Pf*s25 domains, effectively rescuing the anti-*Pf*s25 response. This was true when r*Pf*s25/8 was formulated with Alum, where the anti-*Pf*s25 responses were restored to levels similar to those achieved by unfused r*Pf*s25. The improvement was even more pronounced when r*Pf*s25/8 was formulated with GLA-SE. Here, anti-*Pf*s25 IgG titers were enhanced relative to the modest titers achieved by r*Pf*s25 formulated with GLA-SE. In fact, the 2.5 μg dose of r*Pf*s25/8 formulated with GLA-SE elicited anti-*Pf*s25 responses comparable to those induced by either *Pf*s25-based antigen when formulated with Alum. Importantly, the anti-*Pf*s25 IgG induced by either *Pf*s25-based vaccine demonstrated potent transmission-reducing activity, irrespective of notable differences in IgG subclass profile. Consistent with previous reports [57], the magnitude of the anti-*Pf*s25 response primarily influenced transmission-reducing activity. Mice immunized with r*Pf*s25 formulated with GLA-SE displayed only modest anti-*Pf*s25 IgG responses with little or no functional activity. Together, these results showed that the genetic fusion of *Pf*s25 to the *Pf*MSP8 carrier was required for i) induction of anti-*Pf*s25 responses in the presence of additional immunogenic targets in a multivalent formulation and ii) induction of anti-*Pf*s25 IgG with functional activity in the context of a GLA-SE-based vaccine formulation. Furthermore, we observed potent transmission-reducing activity of vaccine-induced IgG at a concentration of 750 μg/ml, a value 1- to 3-fold lower than the normal level of IgG in mouse serum. These data increase the likelihood that immunization of human subjects with *Pf*s25/8 formulated with GLA-SE can induce functional antibodies that significantly impact parasite transmission if comparable vaccine immunogenicity is achieved.

Initial testing of a bivalent vaccine containing r*Pf*s25/8 and r*Pf*MSP1/8 in outbred mice demonstrated the induction of strong B cell responses against both *Pf*s25 and *Pf*MSP1$_{19}$ that were comparable to those induced by corresponding monovalent vaccines. Anti-*Pf*s25 responses induced by r*Pf*s25/8 were strong and similar regardless of adjuvant selection or the presence of r*Pf*MSP1/8. Interestingly, anti-*Pf*MSP1$_{19}$ responses induced by immunization with

the bivalent vaccine were adjuvant dependent, with the GLA-SE-based formulations eliciting superior responses relative to the Alum-based formulations. This is reflective of several clinical trials in which $rPf$MSP1$_{42}$ formulated with Alum resulted in suboptimal anti-*Pf*MSP1 responses [19, 58]. In addition to the lack of antigenic interference with either adjuvant formulation, these studies also showed that *Pf*MSP8 can be effectively used as a carrier for two distinct vaccine components when administered in the same formulation to genetically heterogeneous, male and female mice.

The systematic evaluation of immune responses generated by the two *Pf*s25-based vaccines as a function of various formulation parameters informed the selection of r*Pf*s25/8 as the more effective candidate. This was most apparent for the induction of transmission-blocking immunity particularly when GLA-SE was selected as adjuvant. In addition, results of the bivalent study suggest that anti-*Pf*MSP1$_{19}$ responses are superior when formulated with GLA-SE, providing some incentive for ultimate selection of this adjuvant. The results of our ongoing comparative immunogenicity studies with inclusion of additional antigens such as *Pf*MSP2 into the multivalent formulation will also impact the choice of adjuvant for advanced testing. With *Pf*MSP2-containing vaccines, we expect that adjuvants such as GLA-SE will be required to effectively induce cytophilic IgG that is needed for opsonization and phagocytosis of merozoites. Overall, these studies further demonstrate the value of *Pf*MSP8 as a carrier protein to help induce effective humoral responses against protective, but poorly immunogenic vaccine components, targeting both blood-stage and sexual stage malaria parasites.

## Supporting information

**S1 Fig. Proliferative responses elicited following stimulation with r*Pf*s25/8 antigen.** CB6F1/J splenocytes (2 x $10^5$/ well) harvested from the indicated immunization groups were stimulated *ex vivo* in triplicate with r*Pf*s25/8 (2 μg/ well) for 96 hrs. [$^3$H]-thymidine (1 μCi/ well) was added for the final 18 hours. Average counts of incorporated [$^3$H]-thymidine were measured for r*Pf*s25/8-stimulated wells and converted into a Stimulation Index (SI) that represents the fold change in proliferation of the indicated condition over the corresponding control wells (media alone). Graphs depict mean SI +/- standard deviation.
(DOCX)

**S2 Fig. Profile of cytokines elicited following stimulation with r*Pf*MSP8.** CB6F1/J splenocytes (5 x $10^5$/well) were harvested from the indicated immunization groups and stimulated *ex vivo* in triplicate with r*Pf*MSP8 (2 μg/well) or in media alone for 96 hours. Culture supernatants were collected and analyzed for production of IL-5, IFNγ, TNFα, -4 and IL-2 using a multiplex assay (Luminex®). To calculate the final concentration of each analyte, the levels found in the corresponding unstimulated conditions were subtracted out as background. Graphs depict mean concentration of each analyte +/- standard deviation.
(DOCX)

**S3 Fig. Anti-r*Pf*s25/8 IgG titers induced by the indicated r*Pf*s25 containing vaccine formulations.** CB6F1/J sera collected 3 weeks following each subcutaneous immunization were analyzed for antigen-specific IgG via ELISA using plates coated with r*Pf*s25/8 (0.25 μg/well). Graphs depict mean IgG titers +/- standard deviation. Asterisks over bars within groups indicate significant boosting of antigen-specific responses over time (Friedman Test; $P < 0.05$ considered significant).
(DOCX)

**S4 Fig. Evaluation of the effect of sex on humoral responses to immunization with combined formulations of r*Pf*s25/8 and r*Pf*MSP1/8 vaccines.** CD1 mice (10/group with 5 male

and 5 female mice) were immunized as indicated and sera were collected following the third immunization. Titers of antigen-specific IgG were measured by ELISA with plates coated with r*Pf*s25 or rGST-*Pf*MSP1$_{19}$ (0.25 ug/well). Graph depicts mean IgG titers +/- standard deviation. Antibody responses in male and female mice within the same immunization group were compared. Statistical significant of differences between sexes were evaluated (Mann-Whitney *U* Test; $P < 0.05$ considered significant; ns, not significant).
(DOCX)

**S1 Table. Cytokine production by cells from mice immunized with *Pf*s25-based vaccines in response to stimulation with Con A.** Splenocytes (5 x 10$^5$/well) were harvested from groups of CB6F1/J mice (n = 5) immunized as indicated and stimulated ex vivo with Con A (0.2 μg/well) or cultured in media alone for 96 hours. Culture supernatants were collected and analyzed for production of IL-5, IFNγ, TNFα, IL-2 and IL-4 using a multiplex assay (Luminex$^®$). To calculate the final concentration of each analyte, the levels found in the corresponding unstimulated conditions were subtracted as background. (ND = not detected).
(DOCX)

**S2 Table. Production of IL-2 and IL-4 by cells from mice immunized with *Pf*s25-based vaccines in response to stimulation with r*Pf*s25/8.** Splenocytes (5 x 10$^5$/well) were harvested from groups of CB6F1/J mice (n = 5) immunized as indicated and stimulated with ex vivo with r*Pf*s25/8 (2 μg/well) or cultured in media alone for 96 hours. Culture supernatants were collected and analyzed for production of IL-2 and IL-4 using a multiplex assay (Luminex$^®$). To calculate the final concentration of each analyte, the levels found in the corresponding unstimulated conditions were subtracted as background. (ND = not detected).
(DOCX)

## Acknowledgments

We thank Dr. Jacqueline Eacret (Drexel University) for critical review of the manuscript.

## Author Contributions

**Conceptualization:** Carole A. Long, James M. Burns, Jr.

**Data curation:** Elizabeth M. Parzych, Kazutoyo Miura.

**Formal analysis:** Elizabeth M. Parzych, Kazutoyo Miura, Carole A. Long, James M. Burns, Jr.

**Funding acquisition:** Carole A. Long, James M. Burns, Jr.

**Investigation:** Elizabeth M. Parzych, Kazutoyo Miura, James M. Burns, Jr.

**Methodology:** Elizabeth M. Parzych, Kazutoyo Miura, James M. Burns, Jr.

**Project administration:** Carole A. Long, James M. Burns, Jr.

**Supervision:** James M. Burns, Jr.

**Writing – original draft:** Elizabeth M. Parzych, James M. Burns, Jr.

**Writing – review & editing:** Kazutoyo Miura, Carole A. Long.

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
