## [Decision Letter · Decision Letter 0]

3 Mar 2020

PONE-D-19-31841

Maintaining immunogenicity of blood stage and sexual stage subunit malaria vaccines when formulated in combination

PLOS ONE

Dear Dr. Burns, Jr.,

Thank you for submitting your manuscript to PLOS ONE. After careful consideration, we feel that it has merit but does not fully meet PLOS ONE’s publication criteria as it currently stands. Therefore, we invite you to submit a revised version of the manuscript that addresses the points raised during the review process.

We would appreciate receiving your revised manuscript by Apr 12 2020 11:59PM. To enhance the reproducibility of your results, we recommend that if applicable you deposit your laboratory protocols in protocols.io, where a protocol can be assigned its own identifier (DOI) such that it can be cited independently in the future. For instructions see: http://journals.plos.org/plosone/s/submission-guidelines#loc-laboratory-protocols

We look forward to receiving your revised manuscript.

Kind regards,

Andrew M. Blagborough

Academic Editor

PLOS ONE

Journal Requirements:

"I have read the journal's policy and the authors of this manuscript have the following competing interests:

JMB is an inventor listed on US Patent No. 7,931,908 entitled “Chimeric MSP-Based Malaria Vaccine”.

4. Please note that all PLOS journals ask authors to adhere to our policies for sharing of data and materials: https://journals.plos.org/plosone/s/data-availability. According to PLOS ONE’s Data Availability policy, we require that the minimal dataset underlying results reported in the submission must be made immediately and freely available at the time of publication. As such, please remove any instances of 'unpublished data' or 'data not shown' in your manuscript and replace these with either the relevant data (in the form of additional figures, tables or descriptive text, as appropriate), a citation to where the data can be found, or remove altogether any statements supported by data not presented in the manuscript.

Reviewers' comments:

Reviewer's Responses to Questions

**Comments to the Author**

1. Is the manuscript technically sound, and do the data support the conclusions?

Reviewer #1: Yes

Reviewer #2: Yes

2. Has the statistical analysis been performed appropriately and rigorously? 

Reviewer #1: Yes

Reviewer #2: Yes

3. Have the authors made all data underlying the findings in their manuscript fully available?

Reviewer #1: Yes

Reviewer #2: Yes

4. Is the manuscript presented in an intelligible fashion and written in standard English?

Reviewer #1: Yes

Reviewer #2: Yes

5. Review Comments to the Author

Reviewer #1: in combination with PfMSP1/8 with the goal that each component maintains their immunogenicity. The goal is to generate a bivalent vaccine designed to target both malaria blood and sexual stages. The authors were successful in achieving this goal.

The introduction was well written.

Figure 1: The authors concluded that both Pfs25 and Pfs28 domains induce antigen-specific T cell response regardless of the vaccine formulation. Whilst they appear not significant, some responses appear higher than they should be (e.g. 2.5 ug dose – anti-PfMSP8 response in the Pfs25-vaccinated mice, anti- Pfs25 response in the PfMSP8-vaccinated mice; and in some other groups – whilst responses in the alum or GLA-SE only groups really showed poor responses). Could the authors comment on this? What could be causing the residual responses? The authors say that graphs depict mean SI +/- SD – how many wells/group? The anti-Pfs25 response seems to be lower in the Pfs25/8 formulation with GLA-SE than the other groups in the 2.5ug dose. Why is this so?

Figure 2. The authors showed cytokine production by CD4+ T cells following vaccination. The methodology indicated that the authors also looked at IL-2 and IL-4, but no data was provided for these cytokines.

Figures 3 and 4. The authors revealed antigen-specific antibodies elicited by Alum- or GLA-SE- formulations. Label should be Anti-PfMSP8 (not PfsMSP-8). The authors are asked to review significant differences between primary, secondary and tertiary vaccinations in anti-PfMSP-8 following Pfs25 vaccination. The background level of antiPfMSP8 antibodies were quite high following Pfs25 vaccination. Why is this so?

All Figures. In figures 3 and 4, the authors started to show comparisons between vaccination groups. Were these comparisons also done with other figures? Please clarify statistical tests - which ones are significant and which ones are not.

The authors are requested to comment on how the outcomes of the membrane feeding assays maybe translatable to human vaccinations.

Reviewer #2: This manuscript presents immunogenicity results in mice when Pfs25 is used in combination with rPfMSP1/8 as a step towards the development of multistage subunit malaria vaccines. In addition, in several experiments authors tested antigen doses (0.5 ug vs 2.5 ug) and two adjuvants (alhydrogel vs GLA-SE). Overall, this is an original research, methods are described in sufficient detail, the study met applicable standards, and conclusions are appropriate. Authors generated a bivalent vaccine, indeed promising and their selected formulation warrants testing in further trials. Interesting is that rPfs25/8-AHG vaccine induced high and comparable titers against Pfs25 irrespective of dose and this was comparable to rPfs25/8-GLA-SE at the highest concentration. GLA-SE influenced more on the IgG subtype. From Table 1, alum adjuvant seems to be more effective than GLA-SE adjuvant, then the preferred formulation with GLA-SE was based on combination with PfMSP1.

I am confused though about the following description in Methods, Lines 141-143: "For antibody analysis, sera were collected three weeks following the first two immunizations and 4 weeks following the final immunization. For T cell studies, mice received a third boost administered by intraperitoneal (i.p.) injection, 8-10 weeks following the final subcutaneous immunization."-- From the description, my understanding is there are 2 vaccinations of 3 week interval, then a third vaccination happened 4 weeks after second dose. For T cell studies, the third dose was administered by ip 8-10 weeks after 2nd vaccine dose? Is this understanding correct and could the authors give additional justification/comment why the third dose would be an ip boost with a different schedule than the rest?

For SMFA, in methods it states that vaccine-induced IgG were mixed at indicated concentrations; in results-- only at one concentration= 750ug/mL. I think then it is best to state the concentration used in the methods section (if indeed only one concentration was used).

In Fig. 9, it was described that strong and similar levels of IgG1 were detected in all groups regardless of antigen formulation or adjuvant but all alum based formulations have very low levels of IgG2a/c and IgG2b-- However, the figure does reflect low levels but not so low. In fact IgG2a/c levels from alum is comparable to that of GLA-SE; IgG2b is lower but almost comparable to IgG1. Only IgG3 was much lower compared to GLA-SE.

minor commend: Discussion: Line 519, "male and female recipients", best to specify male and female mice

6. PLOS authors have the option to publish the peer review history of their article (what does this mean?). If published, this will include your full peer review and any attached files.

Reviewer #1: No

Reviewer #2: No

---

## [Author Response · Author response to Decision Letter 0]

11 Apr 2020

Response to Reviewers

Maintaining immunogenicity of blood stage and sexual stage subunit malaria vaccines when formulated in combination (PONE-D-19-31841)

 Thank you for the comments from the review of our manuscript. We were happy to see that the reviewers felt our studies had merit and that we have the opportunity to submit a revised manuscript that addresses the points raised during the review process. We have addressed journal requirements and the reviewers’ comments as follows:

Journal Requirements.

1. PLOS ONE’S style requirements.

 We have reviewed the PLOS ONE style templates (main body, title page) including those for naming files. We have revised our manuscript throughout to meet these format requirements.

2. Competing Interests Section. 

 We confirm that our stated competing interest does not alter our adherence to all PLOS ONE policies on sharing data and materials. In our cover letter, we have included an update to our competing statement by adding the following: "This does not alter our adherence to PLOS ONE policies on sharing data and materials.” 

3 and 4. Data not shown. 

 In our original manuscript, we had four instances of ‘data not shown’. In the revised manuscript, we have added two supplemental tables (S1 and S2 Tables), updated Fig S2 to include 4 panels of additional data, and included one additional supplemental figure (Fig S4). These provided the relevant data that we originally referenced as ‘not shown’. In the fourth instance, we felt that the data were not a core part of the research being presented in this study and have removed the phrase from the revised document.

Reviewer #1.

1. Figure 1: The authors concluded that both Pfs25 and Pfs28 domains induce antigen-specific T cell response regardless of the vaccine formulation. Whilst they appear not significant, some responses appear higher than they should be (e.g. 2.5 ug dose – anti-PfMSP8 response in the Pfs25-vaccinated mice, anti- Pfs25 response in the PfMSP8-vaccinated mice; and in some other groups – whilst responses in the alum or GLA-SE only groups really showed poor responses). Could the authors comment on this? What could be causing the residual responses? The authors say that graphs depict mean SI +/- SD – how many wells/group? The anti-Pfs25 response seems to be lower in the Pfs25/8 formulation with GLA-SE than the other groups in the 2.5ug dose. Why is this so?

i. The reviewer correctly notes that we see slightly higher than expected proliferation of cells from rPfs25-vaccinated mice when stimulated with rPfMSP8 as well as the reverse, the proliferation of cells from rPfMSP8 vaccinated mice when stimulated with rPfs25. We had commented on similar findings in our antibody data presented in Figure 3 and 4. As these responses are above that observed in adjuvant control, we expect this is due to a shared epitope(s) between rPfs25 and rPfMSP8 associated with a common His-tag leader and linker sequence. Of interest, this cross-reactive response is more prominent in Alum-based formulations versus GLA-SE-based formulations. Importantly, the level of cross-reactive response we observe in these in vitro assays is low in comparison to the potent responses directed against the homologous antigen used for immunization. We have modified both the results further to note these observations. 

ii. For the proliferation data presented in Figure 1, the stimulation indices (+/-SD) were calculated based on five mice per group, with each assay including triplicate wells per mouse and per condition. We have modified the Materials and Methods to clarify this point. 

iii. The reviewer correctly points out that we see a reduced rPfs25-specific proliferative response of cells from mice immunized with 2.5 µg of Pfs25/8 formulated with GLA-SE. We had noted this in our original manuscript and have now modified the text to suggest a possible explanation. We expect that this may be due to some shift of the T cell response toward epitopes in rPfMSP8 at the higher doses but this will require further investigation. Importantly, this potential difference in T cell epitope utilization had no impact on the magnitude or functionality of the antibody responses directed to rPfs25 in rPfs25/8 immunized mice. 

2. Figure 2. The authors showed cytokine production by CD4+ T cells following vaccination. The methodology indicated that the authors also looked at IL-2 and IL-4, but no data was provided for these cytokines.

 These data were originally referenced as ‘data not shown’. We now provide these data in two supplementary tables (S1 and S2 Tables) and in four panels added to S2 Fig. 

3. Figures 3 and 4. The authors revealed antigen-specific antibodies elicited by Alum- or GLA-SE- formulations. Label should be Anti-PfMSP8 (not PfsMSP-8). The authors are asked to review significant differences between primary, secondary and tertiary vaccinations in anti-PfMSP-8 following Pfs25 vaccination. The background level of antiPfMSP8 antibodies were quite high following Pfs25 vaccination. Why is this so?

 The labels have been corrected to indicate ‘anti-PfMSP8’. We have added information on the statistical analysis of the anti-PfMSP8 response following immunization with rPfs25, noting a significant increase in secondary or tertiary immunization. We addressed this observation in our original manuscript as follows: ‘This reactivity is associated with a shared epitope(s) present within the His-tag and linker that are common to both antigens. This reactivity is relatively low, representing only 1-2% of the overall anti-Pfs25 titer induced by immunization with unfused rPfs25 (Fig 3A-B).’

4. All Figures. In figures 3 and 4, the authors started to show comparisons between vaccination groups. Were these comparisons also done with other figures? Please clarify statistical tests - which ones are significant and which ones are not.

 Our studies evaluated three main parameters (antigen, adjuvant, dose) to determine the formulation that induces optimal immune responses upon immunization. We have completed a comprehensive statistical evaluation of immune responses. For clarity of our presentation of the statistical analysis however, we tried to draw attention to comparison regarding specific scientific questions for each set of data deemed most appropriate. For example in the T cell analysis, the focus was on defining the domain-specificity (via proliferation; Fig. 1) and type (Th1 or Th2 via domain-specific cytokine production, Fig 2) of the responding T cells. Here, comparisons of immunized mice versus adjuvant controls within a group were key and were highlighted. However, for the antibody analysis, it was important for us to include some comparison of responses between groups to address the question of competition between antigens when formulated in combination (i.e. Figures 3-5, 7). As noted above, we have included additional statistical analysis in Figure 3 and 4. We have clarified aspects of the statistics in Figure 5. We have added statistical analysis of the IgG1 response in Figure 9. We would be happy to include any additional statistical analysis that reviewers or the editors feel would be meaningful and would improve the manuscript. 

5. The authors are requested to comment on how the outcomes of the membrane feeding assays maybe translatable to human vaccinations.

 We have added the following comment to the Discussion.

 “Furthermore, we observed potent transmission-reducing activity of vaccine-induced IgG at a concentration of 750 µg/ml, a value 1- to 3-fold lower than the normal level of IgG in mouse serum. These data increase the likelihood that immunization of human subjects with Pfs25/8 formulated with GLA-SE can induce functional antibodies that significantly impact parasite transmission if comparable vaccine immunogenicity is achieved.”

Reviewer #2.

1. I am confused though about the following description in Methods, Lines 141-143: "For antibody analysis, sera were collected three weeks following the first two immunizations and 4 weeks following the final immunization. For T cell studies, mice received a third boost administered by intraperitoneal (i.p.) injection, 8-10 weeks following the final subcutaneous immunization."-- From the description, my understanding is there are 2 vaccinations of 3 week interval, then a third vaccination happened 4 weeks after second dose. For T cell studies, the third dose was administered by ip 8-10 weeks after 2nd vaccine dose? Is this understanding correct and could the authors give additional justification/comment why the third dose would be an ip boost with a different schedule than the rest?

 We apologize for the confusion and have modified the Materials and Method to clarify the immunization protocol. For assessment of antibody responses, mice were immunized subcutaneously, three times at 4-week intervals. Sera samples were collected three weeks following the first two immunizations and 4 weeks following the final immunization. For assessment of T cell responses, mice were immunized subcutaneously three times at 4-week intervals. Following an 8-10 week rest, mice received an additional boost administered by intraperitoneal (i.p.) injection to increase trafficking of antigen-specific T cells to the spleen. Splenocytes were harvested 2 weeks following the i.p. boost.

2. For SMFA, in methods it states that vaccine-induced IgG were mixed at indicated concentrations; in results-- only at one concentration= 750ug/mL. I think then it is best to state the concentration used in the methods section (if indeed only one concentration was used).

 We have modified the Materials and Methods to indicate that purified IgG was tested at a concentration of 750 µg/ml. 

3. In Fig. 9, it was described that strong and similar levels of IgG1 were detected in all groups regardless of antigen formulation or adjuvant but all alum based formulations have very low levels of IgG2a/c and IgG2b-- However, the figure does reflect low levels but not so low. In fact IgG2a/c levels from alum is comparable to that of GLA-SE; IgG2b is lower but almost comparable to IgG1. Only IgG3 was much lower compared to GLA-SE.

We thank the reviewer for pointing out a discrepancy between the data presented in Figure 9 and our description of the findings in the Results section with regard to vaccine-induced IgG1 and IgG2a/c levels. We discovered an error in Figure 9. While the description of the data in the text was accurate, the panels for IgG1 and IgG2a/c in Figure 9 were switched. We have corrected this error in the Figure. We show that strong and similar levels of IgG1 were detected in all groups regardless of antigen formulation or adjuvant, we do see difference in vaccine-induced IgG2a/c with significantly higher levels in GLA-SE vs Alum based formulation. We apologize for this error and are happy to have had the opportunity to correct the mistake and this point in the review process.

4. Minor comment: Discussion: Line 519, "male and female recipients", best to specify male and female mice

 The sentence has been change to read ‘male and female mice’.

---

## [Editor Report · Decision Letter 1]

14 Apr 2020

Maintaining immunogenicity of blood stage and sexual stage subunit malaria vaccines when formulated in combination

PONE-D-19-31841R1

Dear Dr. Burns, Jr.,

We are pleased to inform you that your manuscript has been judged scientifically suitable for publication and will be formally accepted for publication once it complies with all outstanding technical requirements.

With kind regards,

Andrew M. Blagborough

Academic Editor

PLOS ONE
---

## [Editor Report · Acceptance letter]

17 Apr 2020

PONE-D-19-31841R1 

Maintaining immunogenicity of blood stage and sexual stage subunit malaria vaccines when formulated in combination 

Dear Dr. Burns, Jr.:

I am pleased to inform you that your manuscript has been deemed suitable for publication in PLOS ONE. Congratulations! Your manuscript is now with our production department. 

With kind regards,

on behalf of

Dr. Andrew M. Blagborough 

Academic Editor

PLOS ONE